# Climate change-induced shifts in survival and size of the worlds' northernmost oviparous snake: A 68-year study

Johan Elmberg[1], Ludvig Palmheden[2], Carl Edelstam[3†], Mattias Hagman[4], Simon Kärvemo[5]*

1 Department of Environmental Science, Kristianstad University, Kristianstad, Sweden, 2 Department of Environmental Monitoring and Research, Swedish Museum of Natural History, Stockholm, Sweden, 3 Department of Zoology, Swedish Museum of Natural History, Stockholm, Sweden, 4 Department of Zoology, Stockholm University, Stockholm, Sweden, 5 Department of Ecology, Swedish University of Agricultural Sciences, Uppsala, Sweden

† Deceased.
* simon.karvemo@slu.se

**Data Availability Statement:** All relevant data are available at: https://datadryad.org/stash/share/0s5gNKywjO_uMg3Xn7SwBl78tEXkSZafbdV_HliIWUA.

## Abstract

Because of their dependence on ambient temperature ectothermic animals can serve as sentinels of conservation problems related to global warming. Reptiles in temperate areas are especially well suited to study such effects, as their annual and daily activity patterns directly depend on ambient temperature. This study is based on annual data spanning 68 years from a fringe population of Grass Snakes (*Natrix natrix*), which is the world's northernmost oviparous (egg-laying) reptile, and known to be constrained by temperature for reproduction, morphology, and behavior. Mark-recapture analyses showed that survival probability was generally higher in males than in females, and that it increased with body length. Body condition (scaled mass index) and body length increased over time, indicative of a longer annual activity period. Monthly survival was generally higher during winter (i.e., hibernation) than over the summer season. Summer survival increased over time, whilst winter survival decreased, especially during recent decades. Winter survival was lower when annual maximum snow depth was less than 15 cm, implying a negative effect of milder winters with less insulating snow cover. Our study demonstrates long-term shifts in body length, body condition and seasonal survival associated with a warming climate. Although the seasonal changes in survival ran in opposite directions and though changes were small in absolute terms, the trends did not cancel out, but total annual survival decreased. We conclude that effects of a warming climate can be diverse and pose a threat for thermophilic species in temperate regions, and that future studies should consider survival change by season, preferably in a long-term approach.

## Introduction

We live in the Anthropocene and in the midst of a global biodiversity crisis. The pace of extinction of species is, if not unprecedented, alarmingly rapid [1, 2]. In addition, among extant

**Funding:** The author(s) received no specific funding for this work.

**Competing interests:** The authors have declared that no competing interests exist.

species there is an undisputed widespread global pattern of population decline, evident in most vertebrate groups [3, 4]. From a global perspective there are many known agents causing species extinction and population decline, for example habitat loss, habitat degradation, over-exploitation, invasive species, and alien pathogens [5]. Global warming is another cause for concern, and one that has received much attention from the scientific community over the last decades. Its effects have dual temporal horizons: 1) the present pace of shifting thermal regimes is too fast to permit selection leading to adaptive evolutionary change, and 2) current populations must respond to changes in ecological time, i.e., within and between generations [6]. How climate change affects survival in current populations of wild animals is therefore central in research on global declines and species extinction.

Ectothermic animals have lower capacity than endothermic to regulate their body temperature and are more dependent on ambient conditions. Consequently, ectotherms are more apt to serve as sentinels of conservation problems related to global warming. Previous research indeed shows that climatic conditions can be a key factor driving intraspecific variation in age-specific mortality (e.g., actuarial senescence, an increase in mortality rate with age) in ectotherms because their metabolism, seasonal activity, and lifespan strongly depend on ambient conditions [7, 8]. There are several studies of ectothermic vertebrates demonstrating that lifespan decreases with increasing temperature [e.g. 9, 10], although there are studies on subtropical reptiles that suggest the contrary [11, 12]. Another feature making terrestrial ectotherms suited for population studies of climate change is that, unlike many mammals and birds, they usually remain in a restricted area throughout life [e.g. 13, 14]. When so, observed patterns and changes of vital rates, life history characteristics, and population change are more likely to depend on local conditions [e.g. 15]. This limits the number of alternative causal explanations, consequently serving to pinpoint crucial implications related to climate change. Finally, ectotherms have a high rate of adaptive phenotypic plasticity, which implies a rapid response to climate change, and this is more pronounced in amphibians and reptiles than in other ectotherms [16].

Second to fishes, reptiles are the most speciose group of ectotherm vertebrates. Approximately 20% of the world's more than 10 000 species of reptiles are in danger of extinction [17], a pattern evident throughout their phylogeny and on all continents. Accordingly, large-scale declines of reptile populations have been observed around the globe [18, 19]. Understanding patterns, and ultimately causality, behind such declines, is obviously of utmost importance. In general terms impact of climate warming on reptiles remains poorly understood [20], but some patterns are evident in previous research. One is that in reptiles, and amphibians, aging, hence mortality rate, is positively related to temperature [10, 21]. An extensive study encompassing 77 species came to a similar conclusion for reptiles, while the opposite trend was observed for amphibians [22]. Accordingly, 29% of the published papers about effects of climate change on reptiles and amphibians found changes in population sizes [23] and climate change may be one of the major drivers of population declines [24].

Reptiles in temperate areas are especially well suited for studying possible effects of climate warming [25, 26]. They have a strong phenotypical response to environmental changes and live in seasonal environments, where annual and daily activity patterns directly depend on ambient temperature. Thus, there is a direct link between climate (change) and the length of the annual activity period, which in turn is set off against life history trait limitations (e.g., annual body growth *versus* energy stores for hibernation, timing of reproduction, and live-bearing reproduction mode *versus* oviparity). The most common measures of changes in body growth are body length and body condition. Both have been shown to change as a result of climate warming across numerous species groups, including reptiles [27].

The projection that higher latitudes will be affected more and faster by climate change than will lower latitudes adds to the urgency of studying effects of global warming in temperate reptiles [28–30]. Temperature is increasing twice as fast in the boreal zone of the northern hemisphere compared to the global average [31]. If reptiles in these areas experience a longer season of activity, they may benefit from increased feeding opportunities and improved body condition, which in turn have positive effects on fecundity [32] and survival [33–36], but see [37]. On the other hand, decreased insulating effect due to reduced snow cover may allow deeper ground freeze and increase winter mortality in reptiles hibernating underground [38]. A longer annual activity period may also increase exposure to predators and parasites, and speed up age-specific mortality [10]. Moreover, northern species at the edge of their distribution commonly exhibit greater sensitivity to environmental changes because of low genetic variability [39].

Understanding changes in abundance trajectories of reptiles inhabiting northern temperate regions facing climate change calls for population level studies of vital rates and traits related to life history. The biggest obstacle for such understanding is the lack of true long-term data relating climate to survival, body condition, and body length, including sex-specific differences thereof. This study is based on a data set from a population of Grass snakes (*Natrix natrix*) collected by the same person over a period of 68 years with consistent methodology in an unchanged protected area. The focal species is the world's northernmost oviparous (egg-laying) reptile, documented to be constrained by climate when it comes to reproduction as well as morphology, behavior, and fitness of hatchlings [40–42], and here studied near the northern limit of its geographic range. Specifically, we use long-term mark-recapture data to answer the following questions: (i) has sex-specific survival probability changed over time? (ii) are climatic variables such as mean temperature, total precipitation, and snow cover associated with such changes?, and (iii) has body condition and body size changed over time and with climatic conditions? Understanding links and responses of this kind is essential for adapting, planning, and executing conservation efforts, as temperatures continue to rise [43].

## Materials and methods

This study is based on data collected by the late Carl Edelstam (Fig 1). From 1942 to 2009 he captured, marked, and recaptured Grass snakes *Natrix natrix* in what is today known as the Nacka nature reserve ('*Nackareservatet*', 7.3 km$^2$, WGS84 dec N 59.29, E 18.15, located in the hemi-boreal biotic zone). The reserve comprises mainly coniferous forest, meadows and pastures, whereas lakes and streams make up 1.04 km$^2$. The data set is probably unique for terrestrial reptiles in its duration and the fact it was collected in a consistent fashion by a single person. Despite the study area being situated within the Stockholm metropolitan area, it has remained in a natural state throughout the study period. Thus, changes observed in the population should not be the result of forestry, ditching, development, or other on-site anthropogenic impact.

We used data from 1,113 grass snake individuals (2,122 capture occasions) caught by Carl Edelstam 1942–2009. For the purpose of this study a total of 868 individuals were considered adults (see below) and sampled mainly at three specific hibernacula on one to three occasions in spring (April to May; N = 660) and autumn (September to October; N = 360). Snakes were captured by hand and kept in cloth bags in the field until examination. Each snake was marked by scale-clipping and its natural color markings were recorded (i.e., the unique ventral pattern, which can be used for individual identification [44]). On each occasion a snake was captured (with few exceptions) date, sex, body length (i.e., snout-vent length, SVL), tail length, and body

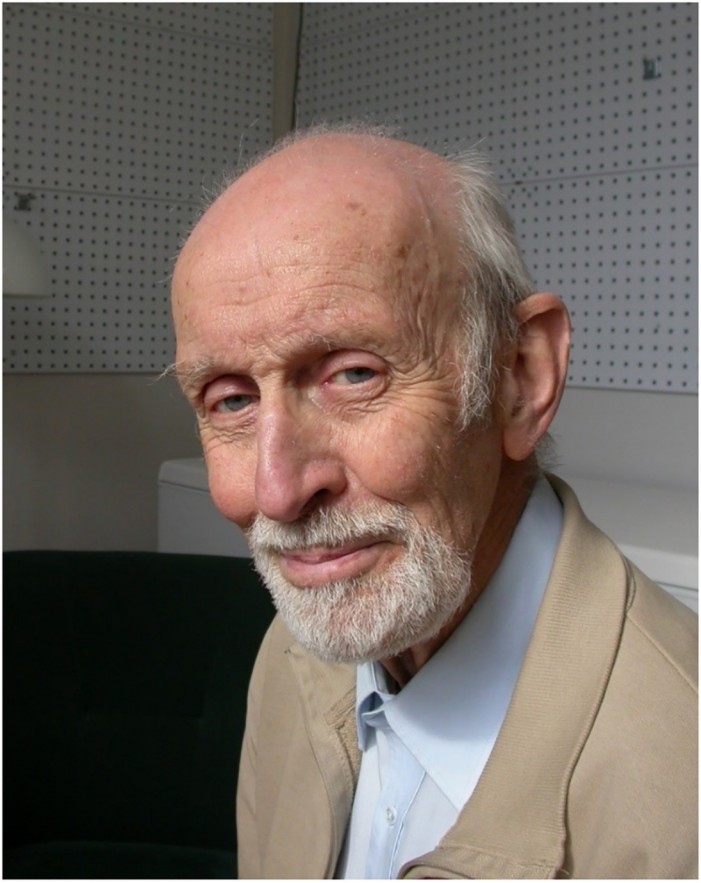

**Fig 1. Carl Edelstam.** This study is based on a 68-year data set from Grass Snakes (*Natrix natrix*) collected by the late Carl Edelstam (1924*-2016†). He was curator of vertebrates at the National Museum of Natural History in Stockholm, Sweden, and received an honorary doctoral degree at Stockholm University. Carl had a life-long devotion for reptiles, but publicly he was known mainly for his work on molt and migration in birds. Photograph from 2008, courtesy of Lars Svensson.

mass were recorded. After examination snakes were returned to the location where they had been captured. The annual sample size was approximately 30 snakes.

Climatic data were obtained from the official weather station Observatoriekullen in Stockholm (WGS84 dec: N 59.341691, E 18.054891), situated 8 km from the study area (Fig 2) and managed by the Swedish Meteorological and Hydrological Institute (SMHI). Data from SMHI were seasonal mean temperature and total precipitation for the annual time periods of June to August ('summer') and November to March ('winter'), to correspond with the subsequent mark-recapture periods in autumn and spring (see below). In addition, data on maximum snow depth and number of days with snow cover were included in the analyses.

## Analyses

To estimate survival probability mark-recapture data were analyzed with the Cormack Jolly-Seber (CJS) model in program MARK [45–47]. Only adult individuals were included in the analyses. Males were considered adult at SVL ≥ 43 cm, and females at SVL ≥ 55 cm (corresponding to total body lengths of 55 and 63 cm, respectively [48]. Based on these premises for

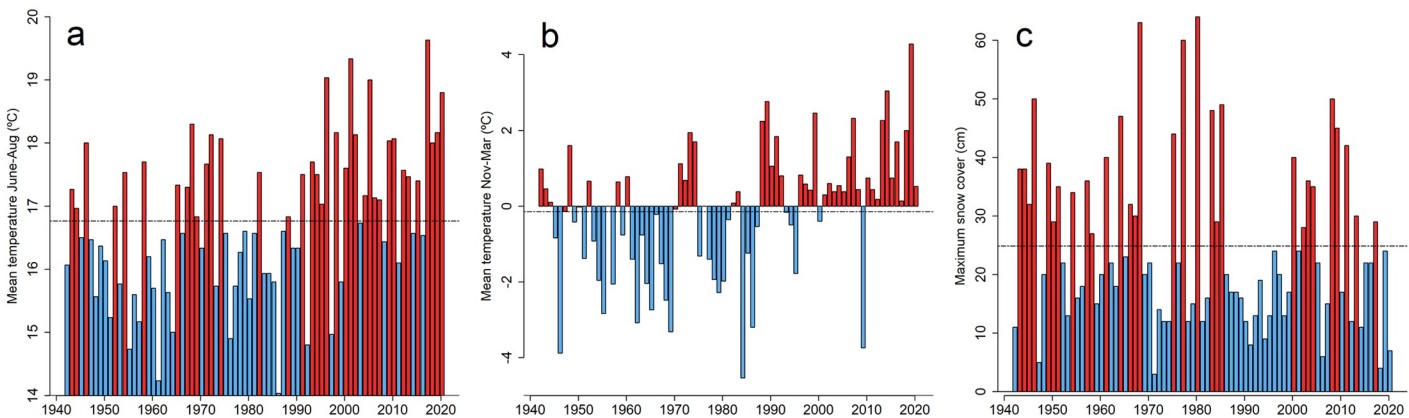

**Fig 2. Temperature and snow cover data.** Climatic data from Observatorielunden in Stockholm from the initiation of the present study in 1942 until 2022, including a) mean summer temperature (June-Aug), b) mean winter temperature (Nov-Mar), and c) maximum snow cover. The dashed lines indicate mean values 1942–2022). Red bars denote years with values above the mean and blue bars year with values below the mean. Source: Swedish Meteorological and Hydrological Institute.

sexual maturity, the best coherent time span of data for survival analysis ranged from autumn 1969 to spring 2002, with only two missed survey occasions (autumns 1980 and 1989), including 359 adult individuals. The Chi-squared two-sample test in program R (ver. 4.0.3 [49]) was used to evaluate the frequency of individual recaptures between 1942 and 2009, as well as between 1969 and 2002.

The long-term mark-recapture history was constructed as a sequence of alternating autumn and spring samples. Consequently, winter survival analyses include the five months from November through March, whereas summer survival analyses include the three months from June through August. The respective apparent survival intervals are therefore 'winter' and 'summer'. This resulted in a data set including 64 capture periods (31 autumns and 33 springs). The CJS model was used to estimate apparent monthly survival (φ; which will be referred to as 'survival' hereafter) and recapture rate (p) [50]. To perform the CJS, model data must meet four assumptions [46, 51]. First, the probability of recapture is the same for every marked animal at a capture event, and second, the probability of survival of a marked individual is the same to the next capture event. These two assumptions were tested and fulfilled by calculating the variance inflation factor ($\hat{c}$) [46]. The other two assumptions; i.e., marks are not lost or missed, and all samples are instantaneous, were assumed to be fulfilled [46, 51].

## Survival across time and seasons

Associations between survival across time and seven biologically relevant parameters for recapture and survival (Table 1) were used to construct a global model.

From the global model we constructed 16 nested models from the variables related to recaptures (Table 1), while the global model parameters related to survival were held constant. These models were analyzed with the CJS model approach. The model with the lowest AICc was selected [52] and compared to competing models by a likelihood ratio test [50]. After the best-fitting model of recaptures had been established it was used to model monthly survival. Similar to when modelling recaptures, nested models constructed from the variables related to survival (Table 1) were compared based on their AICc, and the best competing models were compared with a likelihood ratio test [50].

**Table 1. Variables used in p (recapture) or/and φ (survival) to estimate associations of recapture and survival over time, and the subscripts used for model notation in CJS analysis.**

| Variable | Description | |
| --- | --- | --- |
| sampling rate | Capture effort: the number of sampling occasions each season | $p$ |
| sex | Sex effect | φ, $p$ |
| SVL | Individual covariate: the effect of the first measured SVL of an individual | φ, $p$ |
| season | The effect of season (winter or summer) | φ |
| year | The effect of the number of years since mark-recapture started | φ |
| sex:season | The interactive effect of sex and season | φ |
| year:season | The interactive effect of year and season | φ |

## Survival and climate

An additional CJS model was constructed to explore associations between climate and survival, excluding the variables year and season. Instead, six climate-related variables were added to the global model: seasonal mean temperature and total precipitation for summer and winter, respectively, and maximum snow depth along with the longest period of snow cover each winter (Table 2). The climatic CJS model was developed using the same procedure as in 'Survival estimates across time and season', resulting in 75 different models.

Winter survival in the best fitted model was additionally explored in program R by adding explanatory binary variables based on five maximum snow depth threshold values (10, 15, 20, 25, and 30 cm). Data were analysed in a linear mixed model (LMM), including arcsine-transformed winter survival as the response variable and the five binary snow depth threshold values in separate models. Sex was included as an interaction term with snow depth thresholds, and year was included as a random factor to control for these variations. Normal distribution of residuals was confirmed by Shapiro tests.

## Body size and condition

Body length of all adult snakes captured within the 62-year span 1945–2006 was used to analyse size change over time (and for the body condition formula). Due to sexual dimorphism, males and females were analysed separately [48]. A linear mixed model (LMM) was carried out for each sex (using the R package lme4 [53]), where body length was analysed versus year, with individual as a random effect. Additional LMMs were performed adding the effect of mean temperature and total precipitation for winter and summer, separately. Winter data

**Table 2. Variables used in p (recapture) or/and φ (apparent survival) to estimate the effects of climatic variables, and the subscripts used for model notation in CJS analysis.** See methods for definitions of seasons.

| Variable | Description | |
| --- | --- | --- |
| sampling rate | Capture effort: the number of sampling occasions each season | $p$ |
| sex | Sex effect | φ, $p$ |
| SVL | Individual covariate: the effect of the first measured SVL of an individual | φ, $p$ |
| temp s | Mean temperature of the summer season | φ |
| temp w | Mean temperature of the winter season | φ |
| rain s | Total precipitation (mm) during the summer season | φ |
| rain w | Total precipitation (mm rain or melted snow) during the winter season | φ |
| Snow depth | The maximum snow depth recorded during the winter. | φ |
| Snow days | The number of days for the longest period with continuous snow cover | φ |

were used for snakes caught in spring, whereas summer data were used for individuals caught in autumn. Year and individual were used as random effects.

Body condition was calculated using the scaled mass index (SMI; [54]). Only adult males were used in these analyses, as females potentially have weight biases if gravid. Data from recaptures were excluded to avoid confounding effects (i.e., each individual was included only the first time it was captured). A two-sample t-test was employed to examine variation in SMI between spring and autumn seasons. To assess whether there were changes in body condition over time an LMM was carried out, where SMI was related to year and individual was set as a random effect. Similar to the analyses of SVL, additional analyses included the effect of the climatic variables (i.e., body length in autumn and spring individuals for summer and winter temperatures/precipitation, respectively).

### Ethics statement

Carl Edelstam, an employee of the National Museum of Natural History in Stockholm, initiated and executed the fieldwork, and devised its methods. Due to his passing, his ethical permissions are unavailable. Nevertheless, the Swedish Board of Agriculture and the Ethical Committee granted current ethical approval for Animal Experiments in Stockholm County in 2021, based on the same methods and for the same population (Dnr 11161–2021). Additionally, collection permits for the Nacka nature reserve were issued by the Administrative County Board of Stockholm (522-13116-2021).

## Results

The sex ratio of the adult individuals caught 1942–2009 was male biased with approximately twice as many males than females (69% and 31%, respectively). In the same sample, the proportion of adult males and females recaptured at least once were 51% and 41%, respectively ($X^2$ = 6.64, p = 0.01). Similar percentages were found in the evaluation for the period 1969–2002 used for the mark-recapture analyses: males: 48%, females 41% ($X^2$ = 1.92, p = 0.166). A higher proportion of recapture in males was confirmed by the mark-recapture analyses for 1969–2002, which demonstrate a difference between sexes, with a higher recapture probability in males (β = 0.689, SD = ± 0.461; S1 Table). Additional important variables influencing recapture probability were season (spring; β = 0.704 ± 0.251), sampling rate (β = 0.11 ± 0.03), and SVL (β = 0.036 ± 0.018). See Tables 1 and 2 for variable explanations.

### Survival across time and seasons

Mark-recapture analyses demonstrated a general decline in survival during the 34 year-period, but this effect was strongly dependent on sex, SVL, and season (S2 Table). Survival probability was higher in males than in females (β = 1.498 ± 0.445) and increased with SVL (β = 0.077 ± 0.019). Survival was higher over winter (i.e., hibernation) than over the summer season (β = 3.766 ± 3.633; Fig 3). However, during the last 20 years of the study period, winter survival declined faster (β = -0.117 ± 0.106) than summer survival increased (β = 0.02 ± 0.027; Fig 3).

### Survival and climate

The most influential climatic variable affecting winter survival was total winter precipitation (β = 0.032 ± 0.034; S3 Table). The latter was positively associated with survival in both sexes (Fig 4), and it was particularly influential in small individuals (S2 and S3 Tables). As in the time series model, survival was higher in males than females (β = 1.496 ± 0.46; Fig 4), increasing with SVL in both sexes (β = 0.075 ± 0.02).

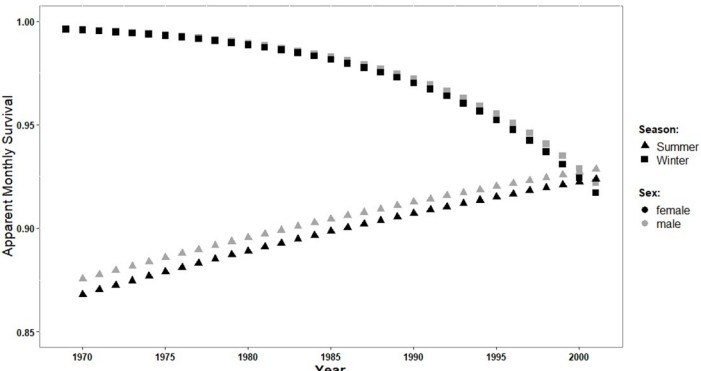

**Fig 3. Survival over time in a population of grass snakes (*Natrix natrix*).** Summer survival (▲) and winter survival (■) across season, and sex (females black symbols, males grey symbols) 1969–2002, estimated from the best-fitted climatic model (S2 Table) and by holding mean body length of females and males, respectively, constant.

The model based on a maximum snow depth above a 15 cm threshold demonstrated a significantly higher winter survival (p = 0.038; Table 3). Survival was higher when maximum snow depth was more than 15 cm and lower when it was less than 15 cm. No other snow depth threshold was associated with such a significant difference.

## Body size and condition

Body condition (SMI) in males was significantly higher in autumn than in spring (t = 4.651, df = 293.3, p < 0.001). Because of this difference, spring and autumn data of body condition were subsequently analysed in separate linear models. SMI increased over the study period; in spring, in autumn as well as when data for both seasons were combined (Table 4). The increase in SMI over time was stronger in individuals caught in autumn than in those caught in spring (Table 4).

Long-term patterns in body length were explored for both sexes but analysed separately. For both and females, a positive association between body length and the consecutive/next study year was found (males: estimate = 0.09, p<0.001; females: estimate = 0.10, p<0.001), i.e., body length increased during the study period in both sexes.

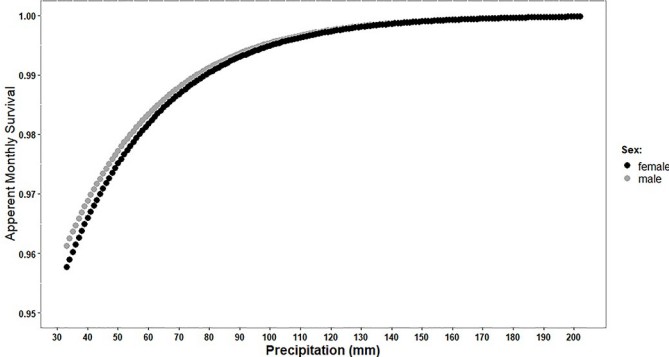

**Fig 4. Survival of grass snakes (*Natrix natrix*) and precipitation.** Winter survival of adult grass snakes in relation to total winter precipitation in females (black) and males (grey) 1969–2002, estimated from the best-fitting climatic model (S3 Table) by holding mean SVL of the respective sex constant.

**Table 3. Model results of the association between winter survival and five snow-depth threshold variables from 1969 to 2002 (binary data including the proportional distribution of data below and over the specific thresholds in 33 winters).**

| Threshold | Estimate | SE | p-value | Below | Over |
|---|---|---|---|---|---|
| >10 cm | 0.021 | 0.022 | 0.348 | 0.09 | 0.91 |
| >15 cm | 0.028 | 0.013 | **0.038** | 0.34 | 0.66 |
| >20 cm | 0.016 | 0.013 | 0.235 | 0.63 | 0.37 |
| >25 cm | 0.012 | 0.015 | 0.413 | 0.75 | 0.25 |
| >30 cm | 0.007 | 0.015 | 0.651 | 0.78 | 0.22 |

**Table 4. Linear mixed model results for body condition (SMI) in adult male grass snakes (*Natrix natrix*) over time 1945–2006, captured in spring, in autumn, and for the two seasons combined.** Significant results are shown in bold.

| Model | Variable(s) | Estimate | Std. Error | df | p-value |
|---|---|---|---|---|---|
| Spring | Intercept | -24.1 | 30.2 | 436.4 | 0.43 |
| | Year | 0.04 | 0.02 | 436.7 | **<0.05** |
| Autumn | Intercept | -93.3 | 46.7 | 228.9 | **<0.05** |
| | Year | 0.07 | 0.02 | 228.9 | **<0.01** |
| Spring+Autumn | Intercept | -88.6 | 27.2 | 511.8 | **<0.01** |
| | Year | 0.1 | 0.01 | 512.2 | **<0.001** |

We also explored associations between climatic variables and body length. Autumn body length was positively related to mean summer temperature the same year (Estimate = 1.3, SE = 0.6, p = 0.04).

## Discussion

We found consistent long-term trends in summer survival (increasing) and winter survival (decreasing). Although the seasonal trends ran in opposite directions and though changes were small in absolute terms, the net effect was a decrease in annual survival. The annual survival rates reported here are somewhat higher than those found for adult grass snakes in southern Sweden and eastern England [33, 55], but the latter study also included immatures, which likely have lower survival. To our knowledge this is the first documentation of long-term survival trends across seasons in a temperate reptile. An obvious implication is that research as well as conservation efforts about effects of climate change on temperate ectotherms need to address seasonal survival patterns separately in order to better understand causality of population change [56, 57]. Teasing apart mortality by season is also crucial for devising countermeasures for declining populations.

Using a temperate ectotherm as a model, our study implies that changed winter climate is far more problematic than changed summer climate. It is alarming that the negative trend in winter survival rather accelerated during the latter decades of the study period. Changes in winter survival and temperature both seem to have intensified starting around 1987 (Figs 2 & 3), coinciding with a 7 cm decrease in the average maximum snow cover before and after that year. If this trend has proceeded until the present day, the current winter survival rate would theoretically be as low as 0.6. Accordingly, our data indicate that the causal driver is not precipitation *per se*, but rather the combination of precipitation and winter temperature (Fig 4, Table 3). In other words, with climate warming much of the winter precipitation in the area falls as rain, which does not protect hibernating snakes [25, 38]. Conversely, lower temperatures and higher overall winter precipitation both increase the likelihood of a 15 cm snow

cover, a condition we observed to increase winter survival (Table 3). Despite rising temperatures and still relatively deep annual snow cover in northern Sweden, there has not been any range expansion northwards over the past 50 years in grass snakes [58].

As far as we know there is no earlier study of a temperate reptile showing such a clear connection of importance for survival during hibernation, let alone one providing a critical threshold for protective snow depth. Sharratt et al. [59] found that a snow depth of 15 cm is when air and soil temperatures stop being strongly correlated. The lack of significance for higher winter survival when snow cover was >20 cm in the present study may be due to more unbalanced data in the models with deeper snow cover (Table 3), which is a long-standing problem in mixed-model inference [53]. The decline in winter survival observed by us may also be attributed to factors such as physiological stress, increased metabolic rates or sudden cold spells when there is no protective snow cover [e.g. 25, 38, 56]. A taxonomically unrelated but illustrative example is blueberries (*Vaccinium sp*.); Wildung & Sargent [60] found a survival probability of <0.20 at snow depths <15 cm, but a survival rate of >0.60 when snow cover was deeper.

Our model predictions suggest that male and female body length increased by 5 and 7 cm, respectively, over the study period 1945–2006. This is line with climate warming permitting a longer annual period of foraging activity and earlier hatching dates [32, 57]. Our data also suggest that grass snakes in the study population now emerge from hibernation approximately 10 days earlier in spring than they did 60 years ago. The resulting longer foraging period is most likely a causative agent also for the trend of improved body condition observed over the study period (cf. [61]). This hypothesis is further supported by the result that the increase in body condition over time was greater in individuals caught in autumn, after summer foraging, than in those caught after hibernation, in spring. Improved body condition at the onset of hibernation should theoretically lead to increased winter survival, but we observed the opposite. Our interpretation is that the negative impact of changed winter climate conditions outweigh the positive effects of increased body condition for hibernation success. We argue that future research on effects of global warming on ectotherms carefully needs to consider such interplay between ecology, physiology, and abiotic conditions across seasons.

We obtained clear answers to all our initial research questions: (i) sex-specific survival probability has changed over time, (ii) climatic variables such as mean temperature, total precipitation, and snow cover are associated with these changes, and (iii) body condition and body size have changed over time and with climatic conditions.

To conclude, we demonstrate profound long-term changes in survival patterns, body condition, body length, and phenology in a grass snake population near its northern range limit. These changes correspond to a long-term trend of milder shorter winters and thus longer annual activity period for ectotherms in the region (Fig 2). Our approach cannot prove causal connections between climate warming and the changes observed in grass snakes, but experimental approaches are hardly possible. Nevertheless, we argue our results provide a very strong case for causality; data were collected by the same person with consistent methods in a study area where climate is the only relevant environmental factor that has changed over the 68-year study period. Moreover, the changes observed in this study population corroborate predictions for temperate zone reptiles in a warming climate; i.e., a longer annual period of activity prolonging the foraging season, leading to faster growth and improved body condition. Yet, the fact that winter survival has decreased consistently, and more rapidly so the last decades, is alarming as it overshadows the observed increase in summer survival. Changed survival patterns like those in this study are a great cause for concern as climate warming continues. They may also spark testable hypotheses about causality for comparative studies addressing increased mortality with increased temperatures [cf. 10]). Future research needs to find out if

such patterns are widespread taxonomically and geographically, and secondly, explore the connection between changed seasonal mortality patterns and population decline. Finally, our findings highlight the extremely high value of long-term data sets like this, collected by Carl Edelstam.

## Supporting information

**S1 Table. Recapture probability.** Details of the five best-fitting CJS models of recapture probability in a population of Grass snakes (*Natrix natrix*), ranked by AICc.
(DOCX)

**S2 Table. Survival over time.** Details of the 10 best-fitting CJS models of survival over time, sex, and season in a population of Grass snakes (*Natrix natrix*), ranked by AICc.
(DOCX)

**S3 Table. Survival associated with climate.** Details of the 10 best-fitting CJS models of survival with climatic data in a population of Grass snakes (*Natrix natrix*), ranked by AICc.
(DOCX)

## Acknowledgments

We sincerely thank Mona Nörklit for donating funds to computerize the extensive original data set. Without her generosity, this study would never have been completed and published. Kristin Löwenborg digitalized the data and Matthew Low assisted in mark-recapture analyses. Nils and Mikael Edelstam provided information about field methods and data coding used by their late father Carl. They also assisted in some of the original field work and read the final manuscript. We thank Lars Svensson for providing the photograph of Carl Edelstam. We have no conflict of interest to declare.

## Author Contributions

**Conceptualization:** Ludvig Palmheden, Carl Edelstam, Simon Kärvemo.

**Data curation:** Johan Elmberg, Carl Edelstam, Mattias Hagman.

**Formal analysis:** Ludvig Palmheden.

**Investigation:** Carl Edelstam.

**Methodology:** Ludvig Palmheden, Carl Edelstam, Simon Kärvemo.

**Project administration:** Simon Kärvemo.

**Resources:** Carl Edelstam.

**Supervision:** Johan Elmberg, Simon Kärvemo.

**Writing – original draft:** Johan Elmberg.

**Writing – review & editing:** Johan Elmberg, Ludvig Palmheden, Mattias Hagman, Simon Kärvemo.

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
