## [Decision Letter · Decision Letter 0]

18 Dec 2023

PONE-D-23-34228Climate change-induced shifts in survival and size of the worlds’ northernmost oviparous snake: a 68-year studyPLOS ONE

Dear Dr. Kärvemo,

Thank you for submitting your manuscript to PLOS ONE. After careful consideration, we feel that it has merit but does not fully meet PLOS ONE’s publication criteria as it currently stands. Therefore, we invite you to submit a revised version of the manuscript that addresses the points raised during the review process. Publishing these long-term data is of great value, I would, however, like to see the original data of Carl Edelstam, not only the processed models. Therefore I urge you to add a table which gives us for each year(season) of observation: no. of snakes recorded, no. of recaptures, size range, and more. 

Please include the following items when submitting your revised manuscript:A rebuttal letter that responds to each point raised by the academic editor and reviewer(s). You should upload this letter as a separate file labeled 'Response to Reviewers'.A marked-up copy of your manuscript that highlights changes made to the original version. You should upload this as a separate file labeled 'Revised Manuscript with Track Changes'.An unmarked version of your revised paper without tracked changes. You should upload this as a separate file labeled 'Manuscript'.If applicable, we recommend that you deposit your laboratory protocols in protocols.io to enhance the reproducibility of your results. Protocols.io assigns your protocol its own identifier (DOI) so that it can be cited independently in the future. For instructions see: https://journals.plos.org/plosone/s/submission-guidelines#loc-laboratory-protocols. Additionally, PLOS ONE offers an option for publishing peer-reviewed Lab Protocol articles, which describe protocols hosted on protocols.io. Read more information on sharing protocols at https://plos.org/protocols?utm_medium=editorial-email&utm_source=authorletters&utm_campaign=protocols.

We look forward to receiving your revised manuscript.

Kind regards,

Ulrich Joger

Academic Editor

PLOS ONE

Journal Requirements:

Additional Editor Comments:

In the section "Survival and Climate" you state that survival was lower when snow cover was less than 15cm. But do I interpret table 3 right that survival was also much lower when snow cover was over 20 cm? Or is there just a weaker correlation with higer snow depths?

I would also like to know whether the records of Natrix natrix in Sweden have increased in the far north during the last decades. Any range extensions?

**Comments to the Author**

1. Is the manuscript technically sound, and do the data support the conclusions?

Reviewer #1: Yes

2. Has the statistical analysis been performed appropriately and rigorously? 

Reviewer #1: Yes

3. Have the authors made all data underlying the findings in their manuscript fully available?

Reviewer #1: Yes

4. Is the manuscript presented in an intelligible fashion and written in standard English?

Reviewer #1: Yes

5. Review Comments to the Author

Reviewer #1: The manuscript addresses an important topic. It is well written and the methodology and results are sound. I have only a few minor recommendations:

Lines 74-76: While this statement is true, I recommend to mention as well that long-distance effects may occur as well. For pollution, effects on penguins are well known but I do not have references available for reptiles. For climate, the following references do show strong such effects:

Grimm-Seyfarth, A., J.-B. Mihoub, B. Gruber & K. Henle (2018). Some like it hot: from individual to population responses of an arboreal arid-zone gecko to local and distant climate. Ecol. Monogr. 88: 336-352.

Grimm-Seyfarth, A., J.-B. Mihoub & K. Henle (2019): Functional traits determine the different effects of prey, predators, and climatic extremes on desert reptiles. Ecosphere 10(9): e02865.

Line 91: See also Reinke et al. (2022) as a recent study that assessed this hypothesis using a very large dataset of aging in reptiles and amphibians.

Reinke, B.A., H. Cayuela et al. (2022): Diverse aging rates in ectothermic tetrapods provide insights for the evolution of aging and longevity. Science 376(6600): 1459-1466.

Evan & Gary (2019) is missing in the reference list. Please cross-check again all citations versus the reference list.

Lines 272-273: Please reverse the 31% and the 69%, as otherwise the statement is confusing.

6. PLOS authors have the option to publish the peer review history of their article (what does this mean?). If published, this will include your full peer review and any attached files.

Reviewer #1: **Yes: **Klaus Henle

---

## [Author Response · Author response to Decision Letter 0]

18 Jan 2024

Response to reviewers

PONE-D-23-34228

Climate change-induced shifts in survival and size of the worlds’ northernmost oviparous snake: a 68-year study PLOS ONE

Dear Dr. Kärvemo,

Thank you for submitting your manuscript to PLOS ONE. After careful consideration, we feel that it has merit but does not fully meet PLOS ONE’s publication criteria as it currently stands. Therefore, we invite you to submit a revised version of the manuscript that addresses the points raised during the review process.

Publishing these long-term data is of great value, I would, however, like to see the original data of Carl Edelstam, not only the processed models. Therefore I urge you to add a table which gives us for each year(season) of observation: no. of snakes recorded, no. of recaptures, size range, and more.

The original data of Carl Edelstam are now uploaded at the Dryad repository including 2122 catches of grass snakes with information about ID-name, Capture occasion, Capture date, Year, Season, Sex, SVL, and Weight. We additionally added Snow cover data and winter survival including Year, Season, Survival, Max snow (cm) and Sex. The data should be found here when published: DOI: 10.5061/dryad.2z34tmpt4

A temporary link is: https://datadryad.org/stash/share/0s5gNKywjO_uMg3Xn7SwBl78tEXkSZafbdV_HliIWUA.

Done

Done

Done

This is ok as it is

No laboratory experiments were conducted in this study

We look forward to receiving your revised manuscript.

Kind regards,

Ulrich Joger

Academic Editor

PLOS ONE

Journal Requirements:

We have changed the manuscript according to the requirements from the links above, including changes of all the text references and the reference list to Vancouver style. 

We have uploaded all data to the Dryad Repository. The link is DOI: 10.5061/dryad.2z34tmpt4. A temporary link to the data is https://datadryad.org/stash/share/0s5gNKywjO_uMg3Xn7SwBl78tEXkSZafbdV_HliIWUA.

As the field worker Carl Edelstam died eight years ago, we have not found his ethical permission. However, we have an ethical permission for the same population and the same methods from 2021, which is now included in the methods. 

The references list and in-text citations have been reviewed and adjusted to adhere to the formatting guidelines of PLOS ONE. 

5. We note that Figure [1] includes an image of a [patient / participant / in the study].

The picture in our manuscript is of neither a patient nor a study object, but it is one of the co-authors. However, he is since long dead and can therefore obviously not provide publication consent. The picture has been chosen in cooperation with two of Carl Edelstam’s sons, and they would be very happy if the picture were to be published in honor of their father’s contribution in this unique study. This issue has additionally been communicated with the Editor (Ulrich Joger) who has consented to our inclusion of the photo in the manuscript.

Additional Editor Comments:

In the section "Survival and Climate" you state that survival was lower when snow cover was less than 15cm. But do I interpret table 3 right that survival was also much lower when snow cover was over 20 cm? Or is there just a weaker correlation with higer snow depths?

Yes, you are correct that the correlations were weaker (and not significant) with higher max snow cover thresholds than 15 cm. Winter survival was also “higher” when snow cover was >20 cm and lower when less than 20 cm” (pos Estimate, but not significant, p=0.235). We only found significant (p<0.05) difference in survival when we used 15 cm as a threshold. This is now clarified in the text. 

“Survival was higher when maximum snow depth was more than 15 cm and lower when it was less than 15 cm“ 

I would also like to know whether the records of Natrix natrix in Sweden have increased in the far north during the last decades. Any range extensions?

There has not been any recent northward range expansion for grass snakes in Sweden. This information is added at Ln 383-385, including a reference from a study of the first author (Elmberg 2023). 

Comments to the Author

1. Is the manuscript technically sound, and do the data support the conclusions?

Reviewer #1: Yes

2. Has the statistical analysis been performed appropriately and rigorously?

Reviewer #1: Yes

3. Have the authors made all data underlying the findings in their manuscript fully available?

Reviewer #1: Yes

4. Is the manuscript presented in an intelligible fashion and written in standard English?

Reviewer #1: Yes

5. Review Comments to the Author

Reviewer #1: The manuscript addresses an important topic. It is well written and the methodology and results are sound. I have only a few minor recommendations:

Lines 74-76: While this statement is true, I recommend to mention as well that long-distance effects may occur as well. For pollution, effects on penguins are well known but I do not have references available for reptiles. For climate, the following references do show strong such effects:

Grimm-Seyfarth, A., J.-B. Mihoub, B. Gruber & K. Henle (2018). Some like it hot: from individual to population responses of an arboreal arid-zone gecko to local and distant climate. Ecol. Monogr. 88: 336-352.

Grimm-Seyfarth, A., J.-B. Mihoub & K. Henle (2019): Functional traits determine the different effects of prey, predators, and climatic extremes on desert reptiles. Ecosphere 10(9): e02865.

Thank you for these suggestions. Regarding long-distance migrations we state in the previous sentence that reptile “usually remain in a restricted area throughout life” and in regard to the studied grass snake population, it is a closed population surrounded by urban areas and long-distance migrations are not possible. Therefore, we kindly ask to not add more information about specific cases of long-distance migration in reptiles. 

However, regarding examples of a positive impact of warmer temperatures on the longevity on reptiles we agree about adding your suggested references. “…although there are studies on subtropical reptiles that suggest the contrary [Grimm-Seyfarth et al. 2018, Grimm-Seyfarth et al. 2019]” at Ln 74.

Line 91: See also Reinke et al. (2022) as a recent study that assessed this hypothesis using a very large dataset of aging in reptiles and amphibians.

Reinke, B.A., H. Cayuela et al. (2022): Diverse aging rates in ectothermic tetrapods provide insights for the evolution of aging and longevity. Science 376(6600): 1459-1466.

Thank you for bringing this significant paper to our attention. Accordingly, we added “An extensive study encompassing 77 species came to a similar conclusion for reptiles, while the opposite trend was observed for amphibians. [Reinke et al 2022]..” Ln 94-95.

Evan & Gary (2019) is missing in the reference list. Please cross-check again all citations versus the reference list.

The citation for Evan & Gary (2019) was already included in the reference list as Cooch EG, White GC, in the program MARK - a Gentle Introduction (19th edn); 2019. It appears that the reference program erroneously used their first names instead of their family names. The reference list has been reviewed once more, and no additional errors were identified. 

Lines 272-273: Please reverse the 31% and the 69%, as otherwise the statement is confusing.

Done

6. PLOS authors have the option to publish the peer review history of their article (what does this mean?). If published, this will include your full peer review and any attached files.

We don’t see the purpose of detailing this history, and do not advocate it. However, we are happy to acknowledge the reviewers and see their names published.

Do you want your identity to be public for this peer review? For information about this choice, including consent withdrawal, please see our Privacy Policy.

Reviewer #1: Yes: Klaus Henle

---

## [Editor Report · Decision Letter 1]

27 Feb 2024

Climate change-induced shifts in survival and size of the worlds’ northernmost oviparous snake: a 68-year study

PONE-D-23-34228R1

Dear Dr. Kärvemo,

We’re pleased to inform you that your manuscript has been judged scientifically suitable for publication and will be formally accepted for publication once it meets all outstanding technical requirements.

Kind regards,

Ulrich Joger

Academic Editor

PLOS ONE
---

## [Editor Report · Acceptance letter]

12 Mar 2024

PONE-D-23-34228R1 

PLOS ONE

Dear Dr. Kärvemo, 

I'm pleased to inform you that your manuscript has been deemed suitable for publication in PLOS ONE. Congratulations! Your manuscript is now being handed over to our production team.

Kind regards, 

on behalf of

Dr. Ulrich Joger 

Academic Editor

PLOS ONE